# OVID: Open Large-Scale Video Dataset as a Novel Source for Image-Text Data

## Abstract

We present OVID, a large open video dataset comprising *10 million hours* of diverse content collected from CommonCrawl. To complement the raw data, we generate image captions for scene-changing frames and video-level captions for a 300M frame–caption subset. Using this subset, we train CLIP models at multiple scales and benchmark them against reference CLIP models trained on DataComp, Re-LAION and DataComp recaptioned with the same captioning pipeline. Observed scaling trends for classification and retrieval show evidence that OVID can be another valuable and scalable source of image-text data, in addition to image-text pairs from public webpages. OVID marks a significant step towards democratizing access to large-scale video data and fostering the development of open multimodal foundation models. To this end, all the data will be freely available to research institutions.

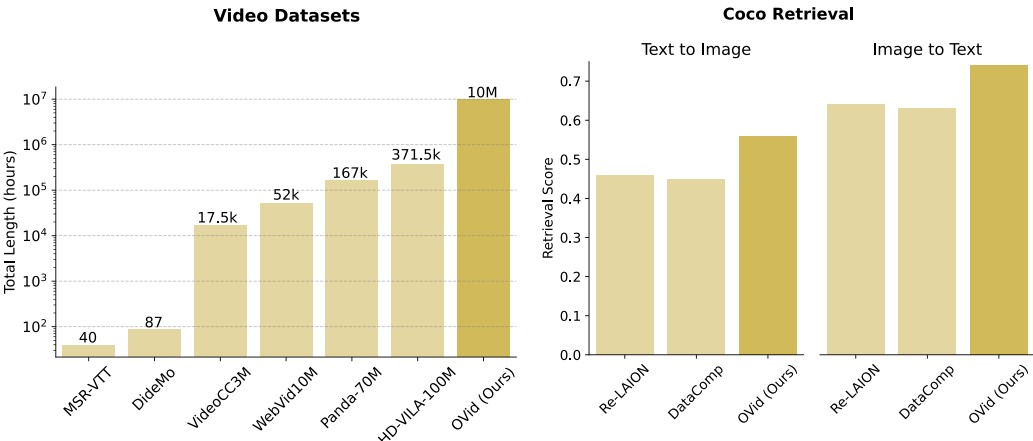

Figure 1: **OVID enables open foundation model scaling at unprecedented scales**. **Left**: With 10M total video hours, OVID is over an order of magnitude larger than existing video-text datasets. **Right**: Our frame-level captions from OVID enable state-of-the-art retrieval on COCO Captions, outperforming Re-LAION and DataComp (more details in Table 5). Since our data is sourced from videos we expect almost no overlap to datasets like Re-LAION and DataComp.

## 1 Introduction

The current paradigm for training capable video or image embedding and foundation models relies on accessible, large-scale datasets. Many publicly available datasets exist for text (Raffel et al., 2020; Gao et al., 2020; Biderman et al., 2022; Penedo et al., 2024; 2023; Weber et al., 2024) and image-text data (Thomee et al., 2016; Sharma et al., 2018; Srinivasan et al., 2021; Changpinyo et al., 2021; Desai et al., 2021; Schuhmann et al., 2021; Byeon et al., 2022; Hu et al., 2022; Wang et al., 2023a; Gadre et al., 2023; Wu et al., 2024b; Li et al., 2024b; Schuhmann et al., 2022). While not quite reaching the scale of proprietary datasets, they are sufficiently large to train competitive open models (Schuhmann et al., 2022; Cherti et al., 2023; Rombach et al., 2022; Huang et al., 2023a). As a prominent example, LAION-5B (Schuhmann et al., 2022) and its 2B English-language subset are today's largest public image-text datasets (with Re-LAION-5B (LAION, 2024) being their recent update) and were used to train popular models like OpenCLIP, KOSMOS-1, and Stable Diffusion (Cherti et al., 2023; Rombach et al., 2022; Huang et al., 2023a).

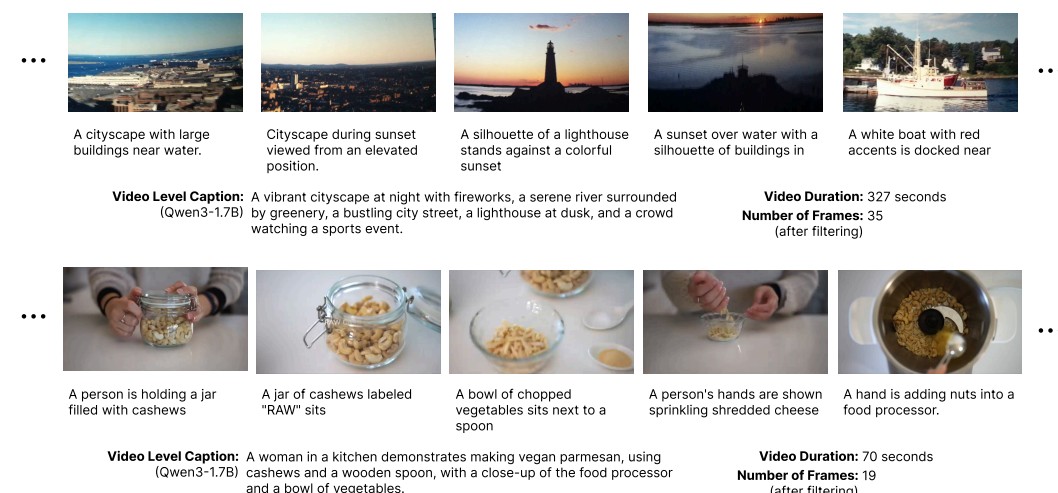

Figure 2: **Example data from 300M image subset of OVID**. We provide a general video caption for each video alongside specific captions for each scene-changing frame.

In contrast, recent open video-text datasets are still relatively small-scale. The largest dataset to date, InternVid (Wang et al., 2023b), contains 234 million video-caption pairs — a relatively meager size compared to modern text and image-text corpora.

The primary bottleneck for large-scale video-text datasets is not the availability of videos per se — millions are accessible on the web — but the significant compute, memory, and engineering effort required to download, filter, and annotate them at scale. Unlike images, videos are often hosted on large commercial platforms that can restrict or gate access, making large-scale collection challenging in practice. Moreover, while smaller-scale video datasets in the past used alt-text and automatic speech recognition (ASR) to produce video captions (Bain et al., 2021; Zellers et al., 2021; Xue et al., 2022), this approach does not scale well. It also produces sub-par video-caption alignment, so that recent methods employ large language models (LLMs) to caption videos based on a combination of frame captions from vision-language models (VLMs) and transcriptions (Wang et al., 2023b; Geng et al., 2024; Chen et al., 2024b; Ju et al., 2024; Xiong et al., 2024).

While other video-text datasets discard generated frame captions, we view the frame-caption pairs generated by our pipeline as an essential component of the final dataset. Existing paired image-text data, including Re-LAION-5B, is almost entirely sourced from internet images. Thus, video frames represent a largely untapped source of image-text pairs that follow a distribution different from existing datasets. We show that our large-scale frame-caption data can be used to train competitive CLIP (Radford et al., 2021) models that yield stronger text-image retrieval performance compared to models trained on existing large-scale image-text data.

The video URLs and captions can be accessed at HuggingFace[1]. Furthermore, research institutions can freely download the raw data upon signing a standard end-user license agreement, which restricts the downloaded data to be used for research purposes.

Overall, we make the following contributions:

---

**Contributions**

- We release **OVID**, a large-scale dataset comprising 1.3B video URLs.
- We make 10M video hours downloaded videos (incl. metadata) freely available to research institutions worldwide for non-commercial use.
- We release 300M high-quality frame-caption pairs as well as 12M video-level text summaries. We show that frame-caption data is a strong and scalable signal for training vision-language models highlighting the data quality of OVID.

---

[1] https://huggingface.co/datasets/EASOJUBYI/urls

| Dataset | Source | Caption Source | English Img-Txt Pairs |
|---|---|---|---|
| MS-COCO (Lin et al., 2014) | Flickr | Manual annotation | 330 k |
| Visual Genome (Krishna et al., 2017b) | MS-COCO + YFCC100M | Manual annotation | 5.4 M |
| YFCC100M (Thomee et al., 2016) | FLickr | Alt-text + Title | 99 M |
| CC3M (Sharma et al., 2018) | Custom web crawl | Alt-text | 3.3 M |
| WIT (Srinivasan et al., 2021) | Wikipedia | Alt-text + Caption | 5.5 M |
| CC12M (Changpinyo et al., 2021) | Custom web crawl | Alt-text | 12 M |
| RedCaps (Desai et al., 2021) | Reddit | Subreddit name + Title | 12 M |
| ALT200M (Hu et al., 2022) | Custom web crawl | Alt-text | 203 M |
| COYO-300M (Byeon et al., 2022) | Common Crawl | Generated | 300 M |
| LAION-400M (Schuhmann et al., 2021) | Common Crawl (Common Crawl) | Alt-text | 413 M |
| COYO-700M (Byeon et al., 2022) | Common Crawl | Alt-text | 747 M |
| DataComp-1B (Gadre et al., 2023) | Common Crawl | Alt-text | 1.4 B |
| LAION-5B (Schuhmann et al., 2022) | Common Crawl | Alt-text | 2.3 B |
| OVID | YouTube, Vimeo, Dailymotion | Generated | 300 M |

Table 1: **OVID compared to publicly accessible image-text datasets**. OVID is the only one sourced from video frames rather than web-crawled images. As a result, it complements existing datasets.

## 2 RELATED WORK

**Multimodal Datasets**. Training multimodal foundation models requires large-scale datasets containing data from at least two modalities.

*Image-text pairs* are easy to collect at scale, since many images on the web are captioned or come with descriptive alt-text. As a result, open image-text datasets have grown rapidly over the past decade (Thomee et al., 2016; Sharma et al., 2018; Srinivasan et al., 2021; Changpinyo et al., 2021; Desai et al., 2021; Schuhmann et al., 2021; Byeon et al., 2022; Hu et al., 2022; Wang et al., 2023a; Gadre et al., 2023; Wu et al., 2024b; Li et al., 2024b), from just 330k English image-text pairs in MS-COCO (Lin et al., 2014) to over 2B in LAION-5B (Schuhmann et al., 2022)'s English-language subset (see Table 1 for more details). Proprietary datasets have been scaled even further to at least 100B image-text pairs (Radford et al., 2021; Jia et al., 2021; Chen et al., 2022; Pham et al., 2023; Peng et al., 2023; Dong et al., 2025; Wang et al., 2025). An overview of non-proprietary image-text datasets is provided in Table 1. However, large image-text datasets see diminishing returns on traditional benchmarks (Wang et al., 2025), and ultimately draw from very similar source distributions. We posit that captioned video frames are a largely untapped source of image-text data.

*Interleaved image-text data* (Alayrac et al., 2022; Zhu et al., 2023b; Laurençon et al., 2023; Huang et al., 2023a; He et al., 2023; McKinzie et al., 2024; Li et al., 2024a; Futeral et al., 2024; Awadalla et al., 2024) can be scaled even further, incorporating more context information at the cost of not-as-well-aligned modalities. In this space, OmniCorpus (Li et al., 2024a) experimented with increasing data diversity by including keyframes and video transcriptions. However, interleaved data is not suitable for all types of models and training recipes.

*Video-text datasets* usually provide captions for clips ranging from three seconds to a few minutes (Wang et al., 2023b; Sun et al., 2024). Many early datasets are specific to domains like movies (Rohrbach et al., 2017; Soldan et al., 2022), cooking (Zhou et al., 2018; Damen et al., 2018), instruction following (Sanabria et al., 2018; Miech et al., 2019), or action recognition (Soomro et al., 2012; Caba Heilbron et al., 2015; Sigurdsson et al., 2016; Kay et al., 2017; Krishna et al., 2017a; Sigurdsson et al., 2018; Goyal et al., 2017; Wang et al., 2019b; Stroud et al., 2020). More relevant for foundation model training are open-domain datasets. Amongst those, smaller dataset can be manually annotated (Xu et al., 2016; Hendricks et al., 2017; Xu et al., 2023a; Liu et al., 2024c), but most recent and larger datasets use automatic speech recognition, subtitles, alt-text, image captions (Zellers et al., 2021; Xue et al., 2022; Bain et al., 2021; Nagrani et al., 2022), and/or utilize LLMs (Wang et al., 2023b; 2024b; Chen et al., 2023b; 2024b; Ju et al., 2024; Xiong et al., 2024; Geng et al., 2024), see Table 2 for more details. An overview of video-text datasets is provided in Table 2. To our knowledge, none of these datasets explore video frames paired with textual captions as a source for image-text data. While InternVid's captioning pipeline involves captioning keyframes (Wang et al., 2023b), these captions are not published or used for model training.

**Vision-Language Foundation Models**. Paired image-text or video-text data is used to train three types of vision-language models (VLMs) (Ghosh et al., 2024) outlined below. OVID's open collection

| Dataset | Source | Captions | Videos | Clips | Duration in h | Median Resolution |
|---|---|---|---|---|---|---|
| MSR-VTT (Xu et al., 2016) | YouTube | Manual | 7.2 k | 10 k | 40 | 240p |
| DideMo (Hendricks et al., 2017) | Flickr | Manual | 10.5 k | 27 k | 87 | – |
| YT-Temporal-180M (Zellers et al., 2021) | YouTube | ASR | 6 M | 180 M | – | – |
| WebVid10M (Bain et al., 2021)* | Stock footage | Alt-text | 10.7 M | 10.7 M | 52 k | 360p |
| HD-VILA-100M (Xue et al., 2022) | YouTube | ASR | 3.3 M | 103 M | 371.5 k | 720p |
| VideoCC3M (Nagrani et al., 2022) | YouTube | Transfer | 6.3 M | 10.3 M | 17.5 k | – |
| Panda-70M (Chen et al., 2024b) | HD-VILA-100M | Generated | 3.8 M | 70.7 M | 167 k | 720p |
| LongVale (Geng et al., 2024) | ACAV-100M (Lee et al., 2021) | Generated | 8.4 k | 105 k | 550 | – |
| LVD-2M (Xiong et al., 2024) | YouTube + Stock footage | Generated | 2 M | 2 M | 11.2 k | 720p |
| InternVid (Wang et al., 2023b) | YouTube | Generated | 7.1 M | 234 M | 760.3 k | 720p |
| MiraData (Ju et al., 2024) | YouTube + Stock footage | Generated | 330 k | 330 k | 16 k | 720p |
| OVID | YouTube, Vimeo, Dailymotion | Generated | 80 M | 2.7 B | 10 M | 720p |

\* this dataset is now defunct

Table 2: **OVID compared to public open-ended video-language datasets**. OVID contains over an order of magnitude more data than the next-largest dataset.

of frame-caption pairs and captioned videos will improve the scale and diversity of training data for all types of VLMs.

*Embedding models* like CLIP (Radford et al., 2021; Ilharco et al., 2021; Cherti et al., 2023), Image-Bind (Girdhar et al., 2023), and other variants (Bao et al., 2022; Xu et al., 2023b; Li et al., 2022b) learn a shared image-text embedding space and are a common component in multimodal models. VideoClip (Xu et al., 2021), VideoMAE (Tong et al., 2022), and ViCLIP (Wang et al., 2023b) are examples of similar embedding models for videos.

*Multimodal LLMs* like Flamingo (Alayrac et al., 2022), BLIP (Dai et al., 2023; Li et al., 2022a; 2023a), LLaVA (Liu et al., 2023; 2024a;d;b; Zhang et al., 2024), many recent GPT variants (Zhu et al., 2023a; Chen et al., 2023a; OpenAI, 2024; Chen et al., 2024a), DeepSeek-VL (Lu et al., 2024; Wu et al., 2024c) and many others (Laurençon et al., 2023; Chen et al., 2022; Peng et al., 2023; Bai et al., 2023; Driess et al., 2023; Piergiovanni et al., 2024; Lin et al., 2023; Luo et al., 2023; You et al., 2023; Wang et al., 2024a) can process images as an input modality, but output only text. Even without an explicit temporal dimension during training, some image-text trained multimodal LLMs exhibit good video understanding (Kim et al., 2024). Other multimodal LLMs like Llama model variants (Zhang et al., 2023; Li et al., 2024c), some GPT variants (Maaz et al., 2023; Su et al., 2023), and others (Zhang et al., 2024; Liu et al., 2024c; Lyu et al., 2023; Yan et al., 2022; Zhao et al., 2023) are specifically trained with video data.

*Large multimodal models* like recent Gemini (Team et al., 2024) models, CoDi (Tang et al., 2023; 2024), Next-GPT (Wu et al., 2024a), and VideoPoet (Kondratyuk et al., 2023) handle images and videos as input and output.

**Image and Video Captioning**. Image and video captioning has a long history in deep learning, both as a benchmark task for visual understanding and as a tool for summarization and abstraction (Vinyals et al., 2016; You et al., 2016; Gu et al., 2017; Sharma et al., 2018; Guo et al., 2020; Sidorov et al., 2020). We refer the reader to Abdar et al. (2024) for an overview.

Automatic captioning pipelines for multimodal data curation at scale commonly follow a shared approach (Wu et al., 2024b; Li et al., 2024b; Wang et al., 2023b; Xue et al., 2024; Chen et al., 2024b; Geng et al., 2024): Images (including the center frame for videos) are captioned using a strong pretrained multimodal LLM. Videos are first split into short clips and might be annotated by an existing video-language model like LLaVA-NeXT-Video (Zhang et al., 2024) or frame-by-frame by a more lightweight model like Tag2Text (Huang et al., 2023b). Audio captions from models like Qwen-Audio / Qwen-Omni (Chu et al., 2023; Xu et al., 2025) or transcriptions from models like Whisper-Large (Radford et al., 2023) can also be incorporated. Final video captions are synthesized from these partial captions by pretrained LLMs like T5 (Raffel et al., 2020), Vicuna (Chiang et al., 2023), Gemini (Team et al., 2024), or Claude (Anthropic, 2024).

## 3 DATASET

In this section, we outline the data collection process for OVID and provide key statistics. Specifically, Section 3.1 details the data curation procedure, Section 3.2 describes the captioning pipeline, and Section 3.3 presents the dataset statistics.

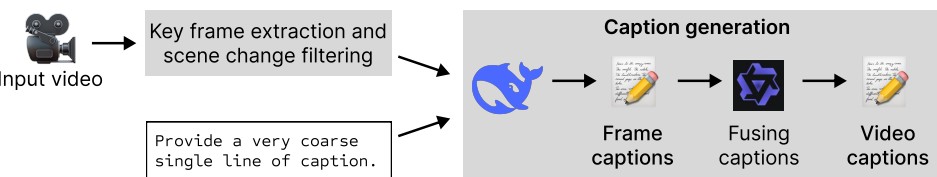

Figure 3: **OVID data curation pipeline**. We source data from Common Crawl Common Crawl and filter for platform-specific video URLs, resulting in 1.3B high-quality video candidates. Out of these, we successfully downloaded 10M video hours using a distributed infrastructure.

### 3.1 DATA CURATION

**Sources**. Large-scale multimodal datasets like LAION-5B (Schuhmann et al., 2021)/Re-LAION-5B (LAION, 2024) and Datacomp-1B (Gadre et al., 2023) rely on Common Crawl as a source of raw data. Following this approach, we use Common Crawl WAT (Web Archive Transformation) files, which provide essential metadata about archived web pages, including HTTP headers and hyperlinks. We extract platform-specific video links using `yt-dlp` (yt dlp, 2021) extractors. To efficiently process this data at scale, we employ the `cc2dataset` (Beaumont, 2022) tool in conjunction with an Apache Spark (Zaharia et al., 2016) cluster. This setup enables the rapid extraction of video URLs and their associated metadata. We use all Common Crawl dumps available as of March 2024, resulting in a corpus of 4.7B candidate URLs. To ensure the quality and accessibility of the videos, we filter for links from major supported platforms – YouTube, Vimeo, and Dailymotion – yielding a final set of 1.3B video URLs.

**Video Download**. We download videos using a distributed setup of 2,000 virtual servers coordinated via a cluster built on Celery and powered by `yt-dlp`. To avoid IP blocking and ensure robust access to video content, we employ residential proxy providers throughout the download process. Overall, our link success rate was approximately 60 %, yielding 10M total video hours.

**Frame Filtering and Moderation**. We extracted keyframes from the videos using `ffmpeg`, and filtered them for black frames. We then extracted scene-changing frames using `ffmpeg`'s scene detection with scene value 0.1. We did not apply additional safety filters at the data collection stage, as our video sources are restricted to well-moderated platforms (YouTube, Vimeo, and Dailymotion). These platforms enforce their own community guidelines and content moderation policies, significantly reducing the prevalence of harmful or unsafe content.

### 3.2 CAPTIONING PIPELINE

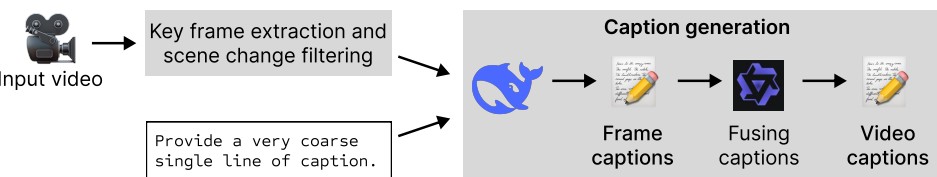

Figure 4: **OVID's captioning pipeline.** Frame-level captions for scene-changing frames are generated using a vision-language model (DeepSeek-VL2-tiny (Wu et al., 2024c)). The resulting annotations are validated through downstream tasks such as zero-shot classification and retrieval. Furthermore, frame captions can be summarized into video-level captions using a language model (Wang et al., 2023b).

Figure 4 illustrates our end-to-end captioning pipeline. We select a VLM that balances quality and efficiency, and then apply it to generate frame-level captions. These captions are optionally summarized into video-level descriptions. We validate the effectiveness of our frame captions through downstream performance on standard benchmarks and explore the impact of caption length on the performance.

**Selecting a Captioning Model**. We considered models for image captioning that balance quality and throughput, selecting the seven top-performing models from the OpenVLM leaderboard (Duan

| Model | Language Model | Vision Model | Params | CLIPScore | Throughput in img/s |
|---|---|---|---|---|---|
| InternVL2.5-1B (Chen et al., 2024c) | Qwen-2.5-0.5B | InternViT-300M-v2.5 | 1 B | 0.43 | **10.41** |
| InternVL2.5-2B-MPO (Chen et al., 2024c) | InternLM2.5-1.8B | InternViT-300M-v2.5 | 2 B | 0.27 | 6.41 |
| InternVL2.5-2B (Chen et al., 2024c) | InternLM2.5-1.8B | InternViT-300M-v2.5 | 2 B | 0.21 | 6.21 |
| SmolVLM-Instruct (Marafioti et al., 2025) | SmolLM2-1.7B | SigLIP-400M | 2.3 B | 0.50 | 2.71 |
| DeepSeek-VL2-tiny (Wu et al., 2024c) | DeepSeekMoE-3B | SigLIP-400M | 3.4 B | **0.62** | 8.20 |
| Qwen2.5-VL-3B-Instruct (Bai et al., 2023) | Qwen2.5-3B | QwenViT | 3.75 B | 0.55 | 2.46 |
| Phi-3.5-vision-instruct (Abdin et al., 2024) | Phi-3.5-mini-instruct | OpenAI CLIP L-14-336 | 4.15 B | 0.59 | 3.74 |

Table 3: **Candidate captioner models**. We choose DeepSeek-VL2-tiny, which achieves the **highest** CLIPScore and the second-highest throughput.

| | Zero-Shot Classification Acc@1 | | | | | | | Img-Retrieval Recall@5 | | |
|---|---|---|---|---|---|---|---|---|---|---|
| | ImageNet-1k | | | ImageNet-R | | ImageNet-Sketch | | COCO Captions | | |
| **Samples** | Original | Recap | Short | Original | Recap | Original | Recap | Original | Recap | Short |
| 12.8 M | 0.10 | 0.09 | 0.10 | 0.12 | 0.15 | 0.04 | 0.06 | 0.10 | 0.21 | 0.19 |
| 30 M | 0.20 | 0.15 | 0.17 | 0.21 | 0.24 | 0.10 | 0.13 | 0.18 | 0.31 | 0.28 |
| 128 M | 0.40 | 0.23 | – | 0.41 | 0.36 | 0.25 | 0.21 | 0.33 | 0.42 | – |

Table 4: **Validation of caption quality**. We report the performance of CLIP-ViT-B-32 trained on subsets of DataComp-1B (Gadre et al., 2023) with their original captions, our automatically generated captions (Recap), and length-constrained captions with only 7 words on average (Short).

et al., 2024) as our starting point. Due to practical constraints on overall compute and GPU memory, we limited our selection to VLMs with fewer than 4B parameters. The resulting candidate pool is provided in Table 3. We use CLIPScore (Hessel et al., 2021) as a proxy to measure the quality of generated captions. Specifically, we calculate CLIPScore using OpenAI CLIP L-14-336 (OpenAI, 2021) on a representative subset of generated captions for 100k images from DataComp-1B Gadre et al. (2023). We select DeepSeek-VL2-tiny for captioning our dataset, as it yielded the highest CLIPScore and second-best throughput. Details for the hyperparameter choices in our pipeline are included in the supplementary material.

**Validation**. To assess caption quality, we recaption subsets of DataComp-1B (Gadre et al., 2023) using DeepSeek-VL2-tiny and train CLIP-ViT-B-32. For validation, we consider the zero-shot image classification performance on ImageNet-1k (Deng et al., 2009), ImageNet-R (Hendrycks et al., 2021), and ImageNet-Sketch (Wang et al., 2019a) alongside image retrieval performance on COCO Captions (Lin et al., 2014). The results are summarized in Table 4. While our automatically generated captions lead to a drop in classification accuracy (36 % for ImageNet, 11 % and 16 % for ImageNet-R and ImageNet-Sketch respectively at the largest data scale), image retrieval performance increases by 27 %. Overall, we find the caption quality acceptable considering the reduced annotation cost and increased dataset scale.

**Caption Length**. We explore different input prompts for captioning and consequently the impact of different resulting caption lengths on downstream model performance. Specifically, we use the following prompt as our default choice: `Provide a very coarse brief single line of caption for the image.` We compare this to using the following prompt, which resulted in caption lengths limited to around 7 words on average (short): `Provide a very coarse brief single line of caption for the main object in the image. Don't worry about the details, just a very high-level description. The description should be as short as possible – 1-3 words.` The impact on model performance of these short captions is also included in Table 4. While we observe a small improvement in classification accuracy on 30.7M training samples, we ultimately decide against artificially constraining the caption length.

**Video-Level Captions**. To generate video-level captions, we aggregate frame-level captions for each video and, following Wang et al. (2023b), we use a language model (Qwen3-1.7B Yang et al. (2025)) to summarize them. This results in concise, high-level descriptions that capture the overall content of

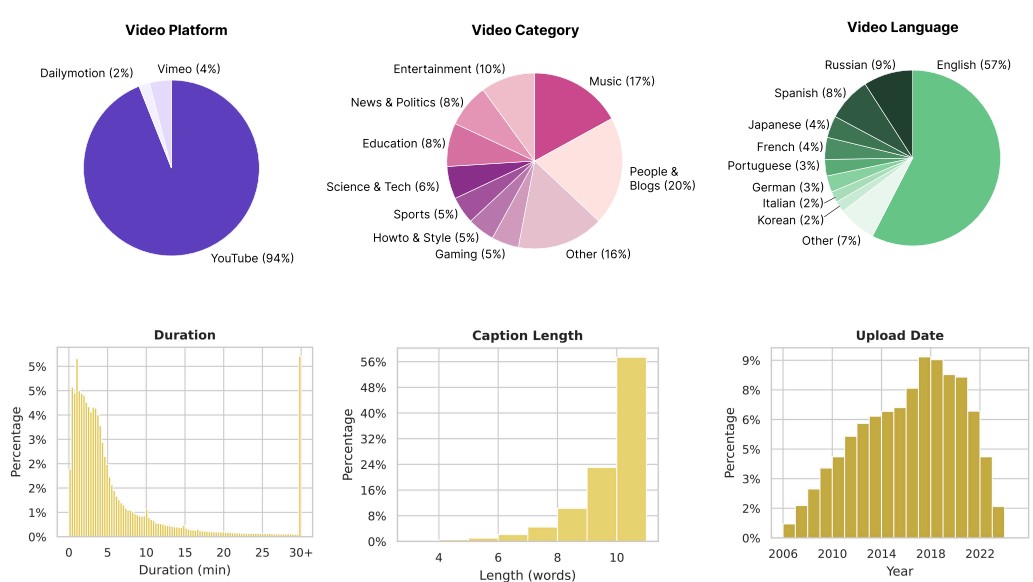

Figure 5: **Dataset statistics for OVID**. **Top row**: Most videos come from YouTube, Vimeo, and Dailymotion, and cover a diverse set of topics. While English remains the most prominent language, nearly half of the videos feature other languages. **Bottom row**: OVID features a substantial fraction of videos exceeding 30 minutes. Our captioning pipeline produces relatively short captions with an average of 9.22 words. Videos cover almost two decades.

each video, as seen in Figure 2. These captions would be useful for training video-text models (e.g. ViCLIP (Wang et al., 2023b)).

## 3.3 DATASET STATISTICS

OVID comprises 10 million video hours, yielding a large corpus of captioned visual content. This amounts to approximately 1 trillion frames, an estimated 2.6 billion of which would be filtered out by our pipeline. On average, we extract and caption 34.5 frames per video. Figure 1 and Tables 1 and 2 contextualize our dataset within existing video-language and image-text pair datasets.

As is the case for other video-language datasets, most videos (over 93 %) are sourced from YouTube. Much smaller fractions (4 % and 2 %) come from Vimeo and Dailymotion.

OVID is exceptionally diverse in its topic coverage. Vlogs (20 %) and music (17 %) are the most common categories, but not by a large margin, resulting in a truly open-ended data distribution. Furthermore, our dataset is noticeably multi-lingual. While the majority (almost 60 %) of video content is in English, many other languages are present in significant proportions.

Like other video datasets, most videos in OVID are below 5 minutes long, with an average duration of 7.82 minutes. However, the distribution is rather long-tailed, and almost 6 % of videos are 30 minutes or longer, supporting long-horizon video tasks.

As mentioned in Section 3.2, our lightweight captioning pipeline is tuned to produce relatively short captions, with most containing between 8-10 words. OVID also presents one of the more recent video datasets in terms of video uploaded, with the most recent entries from early 2024.

To quantify distributional differences between web-image data and our video-derived frames, we compute a Fréchet Inception Distance (FID) over CLIP-ViT-B/32 embeddings using 100k randomly sampled images from each source. We observe a substantial shift between the distributions of ReLAION and OVID (FID = 33.92), while two independent 100k samples from ReLAION yield a near-zero FID (0.16). This confirms that OVID provides a complementary visual distribution to existing web-scale image datasets, supporting its value as an additional pre-training source.

For additional insights, Figure 5 provides a summary of key dataset metrics. A comparison of OVID to existing vision-text datasets can be found in Tables 1 and 2.

| Model | Dataset | Samples | Acc ImageNet-1k | COCO Retrieval Recall@5 Text-to-Image | Image-to-Text | Acc ImageNet-R | ImageNet-Sketch | ImageNet-V2 |
|-------|---------|---------|-----------------|---------------------------------------|---------------|----------------|-----------------|-------------|
| ViT-B-32 | Re-LAION | 30.7M | 0.17 | 0.20 | 0.32 | 0.22 | 0.10 | 0.15 |
| | | 64M | 0.26 | 0.30 | 0.45 | 0.32 | 0.17 | 0.22 |
| | | 128M | 0.35 | 0.38 | 0.53 | 0.40 | 0.24 | 0.28 |
| | | 300M | 0.44 | 0.46 | 0.64 | 0.52 | 0.32 | 0.37 |
| ViT-B-32 | DataComp | 30.7M | 0.20 | 0.21 | 0.32 | 0.25 | 0.12 | 0.17 |
| | | 64M | 0.31 | 0.29 | 0.43 | 0.36 | 0.21 | 0.26 |
| | | 128M | 0.40 | 0.37 | 0.54 | 0.45 | 0.29 | 0.33 |
| | | 300M | **0.50** | 0.45 | 0.63 | **0.57** | **0.39** | **0.42** |
| ViT-B-32 | OVID | 30.7M | 0.08 | 0.27 | 0.40 | 0.13 | 0.04 | 0.08 |
| | | 64M | 0.13 | 0.40 | 0.56 | 0.18 | 0.06 | 0.11 |
| | | 128M | 0.19 | 0.48 | 0.67 | 0.26 | 0.11 | 0.16 |
| | | 300M | 0.24 | **0.56** | **0.74** | 0.34 | 0.16 | 0.21 |
| ViT-B-16 | Re-LAION | 128M | 0.41 | 0.45 | 0.62 | 0.48 | 0.29 | 0.34 |
| | | 300M | 0.51 | 0.53 | 0.71 | 0.59 | 0.38 | 0.44 |
| ViT-B-16 | DataComp | 30.7M | 0.22 | 0.26 | 0.38 | 0.28 | 0.15 | 0.21 |
| | | 64M | 0.38 | 0.36 | 0.52 | 0.42 | 0.26 | 0.32 |
| | | 128M | 0.47 | 0.43 | 0.60 | 0.52 | 0.34 | 0.40 |
| | | 300M | **0.58** | 0.52 | 0.70 | **0.64** | **0.44** | **0.49** |
| ViT-B-16 | OVID | 30.7M | 0.10 | 0.35 | 0.51 | 0.16 | 0.05 | 0.10 |
| | | 64M | 0.18 | 0.50 | 0.67 | 0.24 | 0.09 | 0.15 |
| | | 128M | 0.22 | 0.56 | 0.74 | 0.30 | 0.14 | 0.19 |
| | | 300M | 0.28 | **0.63** | **0.80** | 0.40 | 0.19 | 0.23 |

Table 5: **Zero-shot classification and retrieval performance across datasets and scales**. While OVID achieves the highest performance on the COCO retrieval tasks, it is consistently weaker in ImageNet-1k classification.

# 4 EXPERIMENTS VALIDATING OVID

To validate the quality of extracted video frames and generated captions, we train CLIP models and evaluate them on several downstream tasks. In the following, we describe our experimental setup and the quantitative results and scaling behavior.

## 4.1 EXPERIMENTAL SETUP

We train CLIP models on different dataset sizes (30.7M, 64M, 128M, 300M frames) and model scales (ViT-B/32, ViT-B/16) on both OVID and two reference datasets, namely DataComp-1B and Re-LAION-2B (LAION, 2024). We evaluate each model on two tasks - zero-shot classification on ImageNet-1k and zero-shot image retrieval on MS-COCO. The hyperparameters for our experiments and details about the compute resources used are provided in the supplementary material.

## 4.2 RESULTS

We present the results of our experiments that validate OVID through CLIP training and evaluating its downstream performance in Table 5. OVID shows **superior performance on text-to-image and image-to-text retrieval** tasks while at the same time, **lags behind on ImageNet-1k** zero-shot classification (see Appendix B for further analysis).

We observe comparable performances for CLIP models trained on OVID and models trained on DataComp-1B recaptioned with our captioning pipeline (e.g. 0.23 ImageNet-1k accuracy compared to 0.19 with OVID when training on 128M frames as can be seen in Tables 4 and 5). We hypothesize that the performance gap between the real and synthetic captions for zero-shot classification on ImageNet-1k might be due to the fact that synthetic captions are usually longer and more descriptive than alt-text from the web while ImageNet-1k contains only labels for each image. Similar observations have also been made in prior works (e.g. Li et al. (2023b)). Moreover, synthetically generated captions suffer from limited diversity compared to real ones (Lai et al., 2024). One common approach (Lai et al., 2024) is to mix the synthetic and real captions to increase the diversity, which also suggests great future potential for OVID.

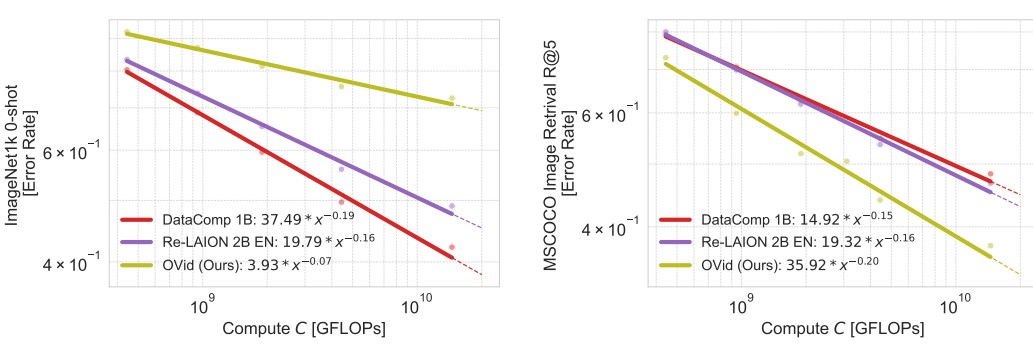

(a) ImageNet-1k zero-shot classification error rate    (b) MS-COCO image retrieval R@5 error rate

Figure 6: **Classification and retrieval scaling trends for OVID, Re-LAION, and DataComp-1B**. OVID has *superior scaling behavior on retrieval* while its *ImageNet zero-shot classification performance falls behind* Re-LAION and DataComp-1B.

In addition to ImageNet-1k, we also evaluate our models on ImageNet-R (Hendrycks et al., 2021), ImageNet-Sketch (Wang et al., 2019a), and ImageNet-V2 (Recht et al., 2019), which further support the generalization capabilities of models trained on OVID.

### 4.3 SCALING TRENDS

The plots in Figure 6 reveal strong scaling trends for OVID across tasks. While all datasets yield improved performance when using larger sets of image-text pairs for training, OVID exhibits weaker scaling for ImageNet-1k zero-shot classification compared to Re-LAION and DataComp. This again confirms that its synthetic captions may be less effective for fine-grained classification. In contrast, OVID demonstrates strong scaling behavior on MS COCO text-to-image retrieval, outperforming other datasets with a steeper slope and lower error rates when using more data. This underscores OVID's effectiveness for retrieval tasks.

## 5 LIMITATIONS

While our video dataset provides a scalable resource for vision-language learning, it has several limitations. Our generated captions introduce a domain gap to real, human-written descriptions. Synthetic captions are often less nuanced and diverse, which can limit generalization capabilities that require rich semantic understanding. Furthermore, we observe relatively low performance for the downstream zero-shot classification task on ImageNet-1k when using OVID. However, this is comparable to the drop in performance when using our captioning pipeline for DataComp-1B data. This confirms that there is a domain gap between our synthetic captions and original captions or alt-text descriptions used for CLIP training.

Moreover, we currently do not filter frames for visual quality or relevance. This inevitably introduces noise into the dataset. Future work could also benefit from principled frame filtering mechanisms (e.g. based on data diversity). Furthermore, we do not consider temporal and audio-visual aspects of the dataset, for instance, by investigating the dataset's impact on training audio-visual video models.

We acknowledge that OVID has undergone only limited curation, since careful curation of such large-scale data would be prohibitively expensive. We rely on content moderation efforts by the video hosting platforms (YouTube, Vimeo, Dailymotion) to prevent harmful content. Nevertheless, models trained on this dataset risk inheriting biases, such as harmful stereotypes.

## 6 DISCUSSION AND CONCLUSION

In this work, we introduce OVID, a large-scale open video dataset featuring 10 million video hours and 300 million frame-caption pairs. Our collection is the largest of its kind, surpassing prior video-language datasets by over an order of magnitude in total video hours.

Our experiments show that CLIP models trained on OVID perform competitively in image and text retrieval tasks, with superior scaling behavior on COCO retrieval benchmarks. At the same time, we

observe a performance gap in zero-shot classification on ImageNet-1k compared to models trained on traditional web alt-text datasets such as DataComp and Re-LAION.

We hope that OVID will democratize access to large-scale video data and spur progress in open multimodal research. All data, including raw videos and metadata will be made freely available to research institutions under a research-only license. By releasing this dataset and establishing a scalable data curation pipeline, we aim to lower the entry barrier for vision-language research and foster reproducibility at scale.

## REPRODUCIBILITY STATEMENT

To ensure reproducibility, we release the full codebase for frame extraction, video-level caption generation, and all preprocessing steps, together with the 1.3B extracted video URLs from CommonCrawl, in the following HuggingFace repository: https://huggingface.co/datasets/EASOJUBYI/urls. The caption annotations for the 300M filtered frames and 12M videos are publicly accessible through this repository. Due to double-blind review constraints, the raw OVID video data (10M video hours) will be made freely available to research institutions under a non-commercial license immediately after acceptance. Upon release, OVID will become the largest open video dataset of its kind, enabling the community to fully reproduce our experiments and advance research on large-scale multimodal language models.

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

# A  TECHNICAL APPENDICES AND SUPPLEMENTARY MATERIAL

## A.1  FRAME FILTERING

In order to obtain high-quality frames for CLIP training, we employ a filtering pipeline as outlined above. Table 6 shows the number of frames extracted and the corresponding percentage of videos.

| Extracted frames | Percentage of videos |
|---|---|
| 1 | 12.93% |
| 2 | 7.96% |
| 3 | 6.31% |
| 4 | 5.36% |
| 5 | 4.74% |
| 6 | 4.21% |
| 7 | 3.79% |
| 8 | 3.44% |
| 9+ | 51.27% |

Table 6: Number of extracted frames per video using our filtering pipeline and corresponding percentage of videos.

## A.2  CAPTIONING

The hyperparameters used for frame-level captioning can be found in Tab. 7

| Hyperparameter | Value |
|---|---|
| Temperature | 0.5 |
| Max Model Length | 4096 |
| Max Sequences | 16 |
| Max Tokens | 20 |

Table 7: Frame-Level Captioning Hyperparameters

To obtain the **video-level** captions, we employed a Qwen3-1.7B (Yang et al., 2025) model with hyperparameters as shown in Table 8 and the following prompt:

```
You are given frame captions of a video.  Your job is to create a
video-level caption.
Just return the video-level caption, nothing else.  Keep it short
and concise but add some details.
Frame captions:
{frame_captions}

\nothink
```

| Hyperparameter | Value |
|---|---|
| Temperature | 0.5 |
| Max Tokens | 512 |

Table 8: Video-Level Captioning Hyperparameters

## A.3  TRAINING DETAILS CLIP MODELS

For training CLIP models we used the OpenCLIP (Ilharco et al., 2021) codebase. We trained models on different datasets with the same setup and hyperpameters outlined in Tab. 9.

| Model | Samples Seen | LR Scheduler | Warmup Steps | LR | GPUs | Batch Size |
|---|---|---|---|---|---|---|
| | 12.8M | cosine | 4000 | 0.001 | 16 GPUs A100 | 2048 |
| | 30.7M | cosine | 3000 | 0.001 | 16 GPUs A100 | 4096 |
| ViT-B/32 | 64.0M | cosine | 4000 | 0.002 | 16 GPUs A100 | 8192 |
| | 128.0M | cosine | 4000 | 0.002 | 64 GPUs A100 | 8192 |
| | 307.2M | cosine | 8000 | 0.002 | 64 GPUs A100 | 16384 |
| | 12.8M | cosine | 4000 | 0.001 | 16 GPUs A100 | 2048 |
| | 30.7M | cosine | 4000 | 0.002 | 16 GPUs A100 | 4096 |
| ViT-B/16 | 64.0M | cosine | 4000 | 0.002 | 16 GPUs A100 | 8192 |
| | 128.0M | cosine | 6000 | 0.002 | 64 GPUs A100 | 8192 |
| | 307.2M | cosine | 4000 | 0.002 | 64 GPUs A100 | 16384 |

Table 9: CLIP scaling laws: training hyperparameters for CLIP ViT-B/32 and ViT-B/16.

## B  WHY STRONG RETRIEVAL DOES NOT TRANSLATE TO IMAGENET-1K ACCURACY

Here we want to expand our analysis of the surprisingly strong retrieval performance (and suboptimal ImageNet-1 classification accuracy) on the filtered OVID captioned frames as shown in Table 5. Table 10 shows that models trained on OVID subsets also achieve strong Flickr30k image-to-text retrieval performance (up to 0.87 R@5).

Additionally, we analyze both the caption length distribution and the coverage of ImageNet-1k class names in two 300M-sample subsets: (i) synthetic captions generated for OVID and (ii) original (alt-text) captions from DataComp.

**Caption length distribution**. Figure 7 shows the frequency distribution of the number of tokens per caption. Captions in the OVID subset exhibit a noticeably *narrower* distribution, with longer captions dominating. In contrast, captions in the DataComp subset display a *broader* distribution, with a substantial fraction of short captions. This indicates that synthetic OVID captions resemble verbose, COCO-style descriptions, while real DataComp captions reflect the more diverse and often concise nature of web text.

**ImageNet-1k class-name coverage**. We further count the occurrences of ImageNet-1k class names (based on the corresponding synsets) in the two subsets. We identify approximately 21M class-name mentions in the OVid captions, compared to 141M in the DataComp captions. Figure 8 and Figure 9 show the top-50 most frequent ImageNet-1k class names for the respective datasets.

This substantial difference in class-name coverage provides a plausible explanation for the weaker ImageNet-1k zero-shot classification performance of models trained on OVID subsets (see Table 10). Real captions from DataComp appear to be more strongly aligned with the semantic structure of ImageNet-1k, containing a far greater density and diversity of class-associated terminology.

Taken together, these findings suggest that caption length or fluency alone is insufficient for improving downstream transfer. Instead, explicit alignment with class-centric semantics – e.g., by increasing the inclusion of ImageNet-1k class names or concept-oriented vocabulary during synthetic caption generation—may be a more effective strategy for boosting zero-shot classification performance on ImageNet-like benchmarks.

## DISCLAIMER FOR USE OF LLMS

LLMs were used exclusively for language polishing and proofreading to improve clarity and readability. All ideas, experimental design, and scientific content originated from the authors.

Table 10: Image-text retrieval performance (R@5) on Flickr30k and MSCOCO for models trained on different 300M-30M subsets of OVID, DataComp, and Relaion. Scaling laws for OVID persist on different models.

| Model | Dataset | Samples | Flickr30k I→T R@5 | MSCOCO I→T R@5 |
|---|---|---|---|---|
| ViT-B-16-text-plus | OVid | 300M | 0.87 | 0.63 |
| ViT-B-16-text-plus | OVid | 128M | 0.82 | 0.56 |
| ViT-B-32 | OVid | 300M | 0.82 | 0.56 |
| ViT-B-16-text-plus | Relaion | 300M | 0.80 | 0.53 |
| ViT-B-16-text-plus | OVid | 64M | 0.79 | 0.50 |
| ViT-B-16-text-plus | DataComp | 300M | 0.75 | 0.52 |
| ViT-B-32 | OVid | 128M | 0.75 | 0.48 |
| ViT-B-32 | Relaion | 300M | 0.72 | 0.46 |
| ViT-B-16-text-plus | Relaion | 128M | 0.71 | 0.45 |
| ViT-B-16-text-plus | OVid | 30M | 0.67 | 0.40 |
| ViT-B-32 | OVid | 64M | 0.67 | 0.40 |
| ViT-B-32 | DataComp | 300M | 0.66 | 0.45 |
| ViT-B-16-text-plus | DataComp | 128M | 0.65 | 0.43 |
| ViT-B-32 | Relaion | 128M | 0.61 | 0.38 |
| ViT-B-32 | DataComp | 128M | 0.55 | 0.37 |
| ViT-B-16-text-plus | DataComp | 64M | 0.55 | 0.36 |
| ViT-B-32 | OVid | 30M | 0.54 | 0.31 |
| ViT-B-32 | Relaion | 64M | 0.51 | 0.30 |
| ViT-B-32 | DataComp | 64M | 0.44 | 0.29 |
| ViT-B-16-text-plus | DataComp | 30M | 0.38 | 0.26 |
| ViT-B-32 | Relaion | 30M | 0.37 | 0.22 |
| ViT-B-32 | DataComp | 30M | 0.32 | 0.21 |

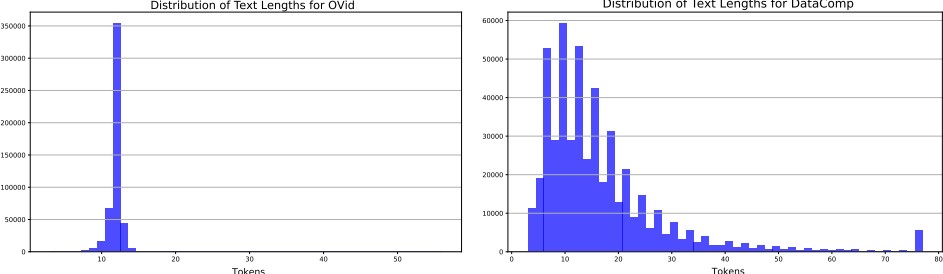

Figure 7: Caption length distribution for the 300M OVid synthetically captioned subset (left) and the 300M DataComp subset (right). OVid captions are longer with a narrower distribution, while DataComp captions show a broader distribution dominated by short captions.

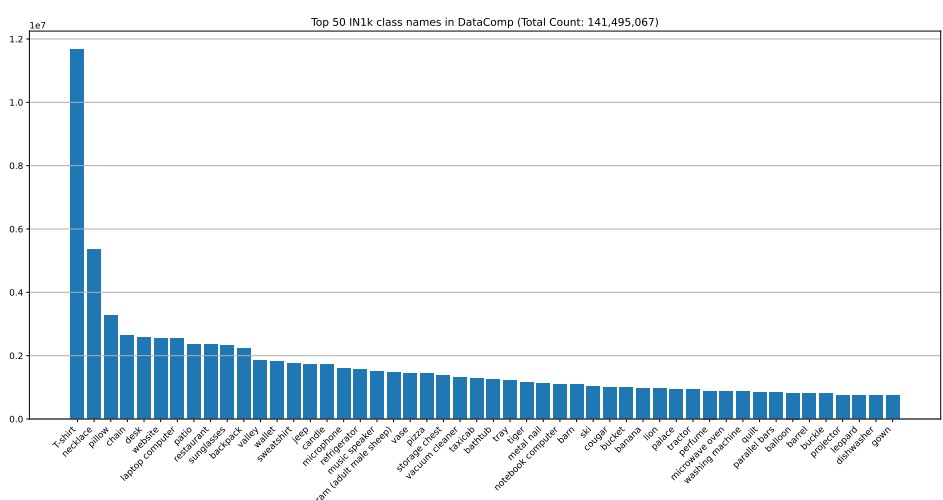

Figure 8: Top-50 most frequent ImageNet-1k class names appearing in the 300M DataComp captions. Real captions from DataComp strongly align with IN1K semantic categories, exhibiting 141M total class-name occurrences.

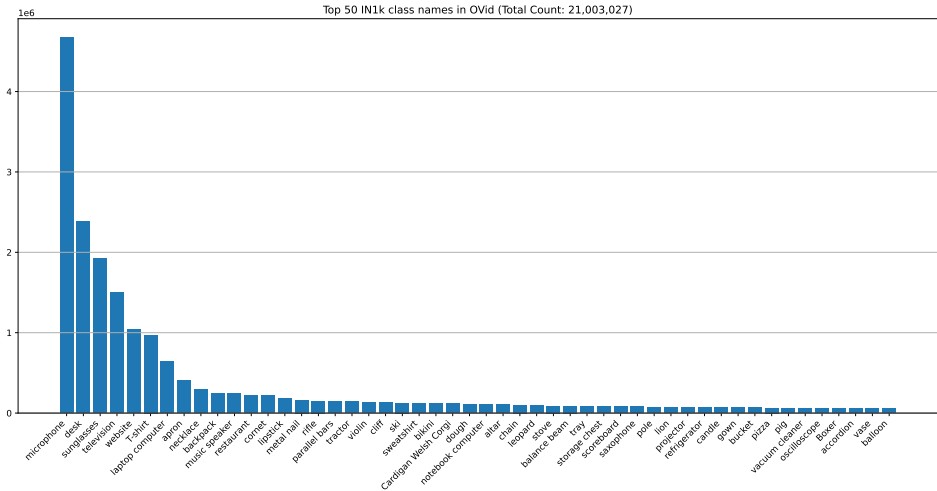

Figure 9: Top-50 most frequent ImageNet-1k class names appearing in the 300M OVid synthetic captions. OVid captions contain 21M total class-name occurrences, substantially fewer than DataComp, consistent with the observed differences in IN1K zero-shot performance.

