# OpenReview forum: "OVid: Open Large-Scale Video Dataset as a Novel Source for Image-Text Data"
_ICLR.cc/2026/Conference — ICLR 2026 Conference Withdrawn Submission_

### Official Review · Reviewer_8Gxi · 2025-10-27

**Soundness:** 2
**Presentation:** 2
**Contribution:** 1
**Rating:** 2
**Confidence:** 4

**Summary:**

This paper introduces a large-scale image-text dataset sourced from 1.3B online videos. From this collection, 300M frame-caption pairs and 12M video-level text summaries are extracted for training vision-language models. By training CLIP models on the newly introduced OVID dataset, the authors demonstrate superior performance on both text-to-video and video-to-text retrieval.

**Strengths:**

1. The OVID dataset collects image-text pairs from large-scale online videos, which is significantly different from previous image-text datasets.

2. Compared to existing open-ended video-language datasets, OVID contains a much larger number of publicly available videos.

**Weaknesses:**

1. I am confused by the statement of contributions in L100–106. It appears that all the contributions focus solely on the release of a large-scale image-text (or frame-text) dataset. In other words, I believe the contributions of this paper are limited, as it primarily presents a dataset.

2. Although OVID is significantly larger than previous datasets, it seems to sacrifice many details in its captions or summaries. As shown in Table 5, CLIP models trained on OVID perform almost the worst on ImageNet-1k, ImageNet-R, ImageNet-Sketch, and ImageNet-V2. The very simple prompt (“Provide a very coarse single line of caption”) used in OVID’s captioning pipeline likely omits too many details from video frames, resulting in unsatisfactory performance on zero-shot classification tasks.

3. In my opinion, the related work section includes many unrelated topics, such as multimodal LLMs and large multimodal models. What is the purpose of connecting these MLLMs to the OVID dataset? Are there any notable similarities or differences between them? The single sentence introducing large multimodal models in  L194–196 seems particularly out of place.

4. I suggest that this dataset paper be resubmitted to a dataset or benchmark track. Meanwhile, the authors may consider how to improve the overall quality of the captions while maintaining the scale of the dataset.

**Questions:**

Please see the weaknesses above.

---

> ### Author Response · Authors · 2025-11-25
> **Rebuttal (Part 1)**
>
> We thank the reviewer for their time and feedback. We will address the reviewers concerns below:
> > 1. I am confused by the statement of contributions in L100–106. It appears that all the contributions focus solely on the release of a large-scale image-text (or frame-text) dataset. …
>
> We apologize for the confusion. Our main contributions are as follows:
> - **1.3B scraped video URLs**, which are already publicly released on Hugging Face (https://huggingface.co/datasets/EASOJUBYI/urls).
> - **A large-scale video dataset of ~10M video hours**, representing a downloaded subset of these URLs. The raw videos cannot be made accessible in an anonymized manner (which would break double-blind constraints), therefore we will make them available free of charge to research institutions following acceptance.
> - **300M frame-captions (and 12M video-level captions)**, which serve as an additional contribution designed to evaluate and demonstrate the quality of the underlying video corpus used for our CLIP experiments. (Released on Hugging Face: https://huggingface.co/datasets/EASOJUBYI/caps_data as well as the captioning pipeline https://huggingface.co/datasets/EASOJUBYI/caps). This constitutes one of many possible dataset usage scenarios.
>
>
> > … In other words, I believe the contributions of this paper are limited, as it primarily presents a dataset.
>
> We would like to emphasize that dataset-focused contributions are fully aligned with the scope of ICLR. The ICLR 2026 Call for Papers explicitly lists “datasets and benchmarks” among the subject areas of interest [1]. We therefore respectfully suggest that the dataset-centric nature of our work should not be viewed as a weakness, but as an appropriate and recognized form of contribution within the venue.
>
> Assembling a 10M-hour multimodal video dataset, validating its quality through large-scale captioning and image-text evaluations, and releasing it for free academic use represents a substantial and novel effort. Large, open datasets of this scale are rare precisely because they require significant engineering, curation, and annotation pipelines. We therefore believe the dataset (and the accompanying analyses) constitute a meaningful and impactful contribution to the community.
>
> [1]https://www.iclr.cc/Conferences/2026/CallForPapers

---

> ### Author Response · Authors · 2025-11-25
> **Rebuttal (Part 2)**
>
> > 2. Although OVID is significantly larger than previous datasets, it seems to sacrifice many details in its captions or summaries. As shown in Table 5, CLIP models trained on OVID perform almost the worst on ImageNet-1k, ImageNet-R, ImageNet-Sketch, and ImageNet-V2. The very simple prompt (“Provide a very coarse single line of caption”) used in OVID’s captioning pipeline likely omits too many details from video frames, resulting in unsatisfactory performance on zero-shot classification tasks.
>
> > 4. Meanwhile, the authors may consider how to improve the overall quality of the captions
>
> While we agree that caption quality can always be improved (e.g., by using larger captioning models), we respectfully disagree that it is a weakness that warrants a rejection of our submission. Our recaptioning pipeline closely follows established practices used in other large video datasets such as InternVid[2]. Prior work (e.g. [3]) has also shown that synthetic captions typically underperform on fine-grained classification tasks but excel in retrieval, which is consistent with our observations. Moreover, the frame-level captions are intended primarily to validate the quality of the underlying data and to demonstrate that OVid can serve, among many use cases, as a new source of image-text pairs with a distribution distinct from standard web images. We further conducted a distributional analysis using CLIP embeddings and found that OVid frames differ substantially from typical web images: the Fréchet Distance between OVid and ReLAION is `33.92`, compared to `0.16` for ReLAION vs. itself. These results, now included in Section 3.3 of the revised manuscript, confirm the significant distribution shift present in OVid.
> Nethertheless, we agree with the reviewer that further investigations into the performance gap between retrieval and classification are necessary. In the revised manuscript, we include such an analysis in Appendix B. Our results show clear differences between synthetic captions in OVid and human-written alt-text in DataComp:
>
>  - **Caption length:** OVid captions exhibit a narrower distribution with consistently longer sentences, whereas human alt-text in DataComp shows a broader distribution dominated by short captions.(https://imgur.com/a/ChenLv8)
> - **Vocabulary and class coverage:** In a 300M subset, synthetic OVid captions contain 21M occurrences of ImageNet-1k class names, compared to 141M in DataComp’s human alt-text. This reduced class-name coverage provides a direct explanation for lower zero-shot IN1k classification accuracy. (see https://imgur.com/a/TOnOQ6L)
>
> These findings confirm that the captioning process, not the underlying image distribution, is the primary driver of the observed performance gap.
> We further assess the OVid-trained CLIP models on Flickr30k retrieval, a setting more aligned with our synthetic captions, and also observe strong performance (see Table 1). Our results show that CLIP models trained on OVid (and its captioning pipeline) achieve strong retrieval performance on COCO and Flickr, tasks that require rich and generalizable visual-textual representations. This provides evidence that the data quality is high. Since these results are obtained with minimal filtering and a small captioning model, they should be interpreted as a lower bound on what the dataset can support.
>
> [2]https://arxiv.org/abs/2307.06942
> [3]https://arxiv.org/pdf/2310.07699
>
> > 3. In my opinion, the related work section includes many unrelated topics, such as multimodal LLMs and large multimodal models. What is the purpose of connecting these MLLMs to the OVID dataset? Are there any notable similarities or differences between them? The single sentence introducing large multimodal models in L194–196 seems particularly out of place.
>
> We included large video and multimodal models in the related work to contextualize the potential usefulness of OVid for future model development. However, we agree that the it feels out of place and will revise that section.

---

> ### Author Response · Authors · 2025-11-25
> **Table 1: Image-text retrieval performance (R@5) on Flickr30k and MSCOCO for models trained on different 300M–30M subsets of OVid, DataComp, and Relaion. Scaling laws for OVid persist on different models.**
>
> | Model                  | Dataset   | Samples | Flickr30k I→T R@5 | MSCOCO I→T R@5 |
> |-----------------------|----------|---------|----------------------|-----------------|
> | ViT-B-16-text-plus    | OVid     | 300M    | 0.87                 | 0.63            |
> | ViT-B-16-text-plus    | OVid     | 128M    | 0.82                 | 0.56            |
> | ViT-B-32              | OVid     | 300M    | 0.82                 | 0.56            |
> | ViT-B-16-text-plus    | Relaion  | 300M    | 0.80                 | 0.53            |
> | ViT-B-16-text-plus    | OVid     | 64M     | 0.79                 | 0.50            |
> | ViT-B-16-text-plus    | DataComp | 300M    | 0.75                 | 0.52            |
> | ViT-B-32              | OVid     | 128M    | 0.75                 | 0.48            |
> | ViT-B-32              | Relaion  | 300M    | 0.72                 | 0.46            |
> | ViT-B-16-text-plus    | Relaion  | 128M    | 0.71                 | 0.45            |
> | ViT-B-16-text-plus    | OVid     | 30M     | 0.67                 | 0.40            |
> | ViT-B-32              | OVid     | 64M     | 0.67                 | 0.40            |
> | ViT-B-32              | DataComp | 300M    | 0.66                 | 0.45            |
> | ViT-B-16-text-plus    | DataComp | 128M    | 0.65                 | 0.43            |
> | ViT-B-32              | Relaion  | 128M    | 0.61                 | 0.38            |
> | ViT-B-32              | DataComp | 128M    | 0.55                 | 0.37            |
> | ViT-B-16-text-plus    | DataComp | 64M     | 0.55                 | 0.36            |
> | ViT-B-32              | OVid     | 30M     | 0.54                 | 0.31            |
> | ViT-B-32              | Relaion  | 64M     | 0.51                 | 0.30            |
> | ViT-B-32              | DataComp | 64M     | 0.44                 | 0.29            |
> | ViT-B-16-text-plus    | DataComp | 30M     | 0.38                 | 0.26            |
> | ViT-B-32              | Relaion  | 30M     | 0.37                 | 0.22            |
> | ViT-B-32              | DataComp | 30M     | 0.32                 | 0.21            |

---

> > ### Comment · Reviewer_8Gxi · 2025-11-27
> >
> > Thank you to the authors for your detailed responses. I would like to provide some further comments regarding your replies:
> >
> > 1. I appreciate the significant effort the authors have invested in collecting and constructing the large OVID dataset. However, in my view, dividing this contribution into three separate points is unnecessary, as they all pertain to the release of a dataset. The inclusion of video URLs and frame-caption pairs should be considered integral components of the dataset. Ultimately, these aspects represent a single contribution. You may claim some differences compared to other image-text datasets (e.g., your data is collected from videos) or discuss the unique advantages offered by OVID inn contributions. However, at present, you don't.
> >
> > 2. Regarding the extremely low performance on zero-shot classification, your current explanations are not convincing. While caption length and class coverage may contribute to poor zero-shot results, these factors should have been considered during your dataset collection and curation.
> >
> > Furthermore, since you are collecting frame-captions from video sources and include a comparison of video datasets in Figure 1, it is surprising that you do not conduct downstream video experiments after training the CLIP model with OVID. In contrast, the original CLIP paper includes such experiments (e.g., UCF101, Kinetics).
> >
> > 3. Overall, as you have acknowledged, the writing in your current paper still requires improvement.

---

> > > ### Author Response · Authors · 2025-12-03
> > >
> > > We thank the reviewer for their additional comments and take the opportunity to clarify the points raised:
> > > 1. We thank the reviewer for their suggestion to rephrase our contributions, we have updated the final paragraph of our introduction (see below).
> > > 2. We strengthened our explanation of the ImageNet zero-shot classification results and clarified that they are consistent with prior research [1]. Optimizing captions for ImageNet zero-shot classification does not necessarily result in generally improved caption quality and arguably defeats the purpose of zero-shot evaluation.
> > > 3. We thank the reviewer for the suggestions to improve the writing. We view these as minor adjustments that we will incorporate in the revision.
> > >
> > > [1] https://arxiv.org/abs/2310.07699
> > >
> > > ---
> > >
> > > ---
> > >
> > > The main elements of our contribution can be summarized as follows:
> > > - We collect and release OVid, a large-scale dataset comprising 1.3B video URLs and make 10M video hours (incl. metadata) freely available to research institutions worldwide for non-commercial use. We provide additional metadata and features to support data curation and filtering. Furthermore, we release 300M high-quality frame-caption pairs as well as 12M video-level text summaries.
> > > - We show that frame-caption data provides a strong signal for training vision-language models that is complementary to existing image-text datasets. We evaluated this on a standard set of text-image zero-shot and retrieval benchmarks.

---

### Official Review · Reviewer_mjz7 · 2025-10-28

**Soundness:** 3
**Presentation:** 3
**Contribution:** 3
**Rating:** 8
**Confidence:** 3

**Summary:**

This paper introduces OVID, a massive open video dataset sourced from Common Crawl, containing 1.3B video URLs (∼10M hours). From these, the authors generate 300M high-quality frame-caption pairs by extracting scene-change frames and captioning them with a efficient vision-language model (DeepSeek-VL2), followed by video-level summarization.

When used to train CLIP models, OVID demonstrates strong and scalable performance: it achieves state-of-the-art results on COCO image-text retrieval, outperforming comparable datasets like DataComp. However, a noted limitation is its weaker performance on zero-shot ImageNet classification, attributed to a domain gap from its synthetic captions.

The work provides a valuable, large-scale multimodal resource complementary to existing image-text collections.

**Strengths:**

This paper's main contribution is OVID, an extremely large-scale video-text dataset (1.3B URLs, 10M hours) that significantly dwarfs existing open collections. Its scale and diversity—multilingual and multi-topic—are its key strengths. The method of generating image-text pairs from scene-change frames is both novel and impactful.

The empirical validation is robust: CLIP models trained on OVID achieve state-of-the-art COCO retrieval performance across scales, proving the data's tangible value. This focus on large-scale, open multimodal data, combined with its commitment to reproducibility and open access, makes it a timely and valuable contribution to the community.

**Weaknesses:**

While the OVID dataset is a significant contribution, several limitations should be noted.

1. Its reliance on synthetic captions introduces a domain gap; while this benefits retrieval tasks, it leads to a notable drop in zero-shot classification accuracy.

2. The light data filtering strategy, while enabling scale, leaves potential concerns about noise, bias, and unsafe content unquantified in the current evaluation.

3. The empirical study is thorough but narrowly focused on image-text tasks, leaving its value for video-language modeling an open question for future work.

**Questions:**

1. The paper notes a performance drop in ImageNet classification due to synthetic captions. It would be helpful to see a quantitative analysis of caption diversity (e.g., vocabulary size, sentence length) compared to human-written alt-text. Furthermore, could the authors comment on whether mixing OVID's data with smaller human-curated subsets might help bridge this domain gap and improve classification accuracy?

2. The reliance on platform-level moderation is noted. Have the authors conducted any evaluation for unsafe or biased content? Even a small-scale audit or analysis of potential demographic biases or NSFW frames would significantly strengthen the claims about dataset reliability and safety.

3. Could the authors elaborate on how they ensure the selected scene-change frames are semantically meaningful and not overly redundant? Would an adaptive, diversity-based sampling strategy be a worthwhile future direction to improve data efficiency?

4. Given the video origins of the data, has there been any consideration to benchmark OVID on temporal or video-language tasks? Demonstrating performance on benchmarks like MSR-VTT could further validate its value for video understanding, beyond image-text retrieval.

5. The dataset uses a research-only license. Could the authors clarify if there will be restrictions on creating derivative datasets or on fine-tuning commercial models? This clarity is important for the community to understand the full implications of the "open access" terms.

6. Figure 6 indicates different scaling trends for retrieval and classification. Could the authors provide more insight into why OVID exhibits stronger scaling for retrieval but weaker scaling for classification? Is this primarily attributed to caption style or domain bias?

7. Do the authors plan to extend OVID with audio or multimodal features? Integrating audio captions could make it an even more valuable resource for training unified vision-language-audio models.

---

> ### Author Response · Authors · 2025-11-25
> **Rebuttal (Part 1)**
>
> We thank the reviewer for their time and constructive feedback. We appreciate that the reviewer thinks our validation is robust and impactful. We will address the reviewers concerns below:
>
> > 6. Figure 6 indicates different scaling trends for retrieval and classification. Could the authors provide more insight into why OVID exhibits stronger scaling for retrieval but weaker scaling for classification? Is this primarily attributed to caption style or domain bias?
>
> > 1. The paper notes a performance drop in ImageNet classification due to synthetic captions. It would be helpful to see a quantitative analysis of caption diversity (e.g., vocabulary size, sentence length) compared to human-written alt-text. Furthermore, could the authors comment on whether mixing OVID's data with smaller human-curated subsets might help bridge this domain gap and improve classification accuracy?
>
> We agree that the performance drop on ImageNet classification warrants a deeper quantitative analysis. We analyzed the caption-length distributions and the frequency of ImageNet-1k (IN1k) class names in the 300M Ovid-recaptioned subset and the 300M DataComp subset. Our analysis shows that Ovid captions have a much narrower length distribution, with longer captions dominating, whereas DataComp captions exhibit a broader distribution with a clear prevalence of short captions (see https://imgur.com/a/ChenLv8). More importantly, we observe only 21M IN1k class-name occurrences in the Ovid captions, compared to 141M in DataComp (see https://imgur.com/a/TOnOQ6L). This gap provides a clear explanation for the weaker IN1k zero-shot classification performance of Ovid-trained models: real captions in DataComp align much more strongly with the IN1k class structure. Consequently, adapting the caption-generation pipeline to better capture IN1k class names is likely to improve performance on IN1k classification. We further assess the OVid-trained CLIP models on Flickr30k retrieval, a setting more aligned with our synthetic captions, and also observe strong performance (see Table 1). We have added this analysis to Appendix B in the revised manuscript.
>
> Regarding the reviewer’s second point, we agree that mixing OVid with smaller human-curated subsets could help mitigate this gap. Our comparison with recaptioned DataComp shows that synthetic captions consistently reduce IN1k classification performance even on datasets known to perform well in their original form. This suggests that incorporating a modest amount of human-curated captions or adapting synthetic captioning to explicitly include more class-relevant vocabulary, could help restore fine-grained classification accuracy.
>
> > 2. The reliance on platform-level moderation is noted. Have the authors conducted any evaluation for unsafe or biased content? Even a small-scale audit or analysis of potential demographic biases or NSFW frames would significantly strengthen the claims about dataset reliability and safety.
>
> We conducted small-scale audits of randomly sampled videos and did not encounter NSFW content in these inspections. Because the dataset relies on platform-level moderation (e.g., YouTube), which typically employs commercial-grade safety filters that are substantially more effective than currently available open-source alternatives, we did not apply an additional NSFW filtering stage.
>
> > 3. Could the authors elaborate on how they ensure the selected scene-change frames are semantically meaningful and not overly redundant? Would an adaptive, diversity-based sampling strategy be a worthwhile future direction to improve data efficiency?
>
> This is an excellent question. In our current release, the only filtering applied is black-frame detection and scene-change detection; we intentionally do not apply additional semantic or diversity-based filtering. Our goal is to provide a simple, transparent baseline that avoids injecting task-specific biases and serves as a lower bound on the dataset quality.
> That said, we fully agree that adaptive or diversity-aware sampling strategies could further improve data efficiency and frame quality. We see these as valuable directions for future work and expect downstream users to adopt such techniques for task-specific curation.

---

> ### Author Response · Authors · 2025-11-25
> **Rebuttal (Part 2)**
>
> > 4. Given the video origins of the data, has there been any consideration to benchmark OVID on temporal or video-language tasks? Demonstrating performance on benchmarks like MSR-VTT could further validate its value for video understanding, beyond image-text retrieval.
>
> We fully agree that evaluating on video-specific tasks (e.g., action recognition, video QA, or training a video LVLM) is an important next step. However, training competitive video models at the scale of OVid requires substantial compute resources, which were beyond the compute budget scope for this work. Our goal with this paper is to release a scalable dataset and provide initial signals of its utility via image-text retrieval and CLIP training. We hope our work lowers the entry barrier and enables the community to explore video-specific training and evaluation.
>
> > 5. The dataset uses a research-only license. Could the authors clarify if there will be restrictions on creating derivative datasets or on fine-tuning commercial models? This clarity is important for the community to understand the full implications of the "open access" terms.
>
> The URLs as well as the code will be MIT license, access to the data will only be for research institutions for non-commercial use.
>
> > 7. Do the authors plan to extend OVID with audio or multimodal features? Integrating audio captions could make it an even more valuable resource for training unified vision-language-audio models.
>
> This is an excellent suggestion. We view the dataset as a foundation for substantial follow-up work, and extending it with additional multimodal signals, such as audio captions or richer audio embeddings, is indeed a promising next step. While such extensions are beyond the scope of the current submission, we plan to continue improving the dataset in future releases.

---

> ### Author Response · Authors · 2025-11-25
> **Table 1: Image-text retrieval performance (R@5) on Flickr30k and MSCOCO for models trained on different 300M–30M subsets of OVid, DataComp, and Relaion. Scaling laws for OVid persist on different models.**
>
> | Model                  | Dataset   | Samples | Flickr30k I→T R@5 | MSCOCO I→T R@5 |
> |-----------------------|----------|---------|----------------------|-----------------|
> | ViT-B-16-text-plus    | OVid     | 300M    | 0.87                 | 0.63            |
> | ViT-B-16-text-plus    | OVid     | 128M    | 0.82                 | 0.56            |
> | ViT-B-32              | OVid     | 300M    | 0.82                 | 0.56            |
> | ViT-B-16-text-plus    | Relaion  | 300M    | 0.80                 | 0.53            |
> | ViT-B-16-text-plus    | OVid     | 64M     | 0.79                 | 0.50            |
> | ViT-B-16-text-plus    | DataComp | 300M    | 0.75                 | 0.52            |
> | ViT-B-32              | OVid     | 128M    | 0.75                 | 0.48            |
> | ViT-B-32              | Relaion  | 300M    | 0.72                 | 0.46            |
> | ViT-B-16-text-plus    | Relaion  | 128M    | 0.71                 | 0.45            |
> | ViT-B-16-text-plus    | OVid     | 30M     | 0.67                 | 0.40            |
> | ViT-B-32              | OVid     | 64M     | 0.67                 | 0.40            |
> | ViT-B-32              | DataComp | 300M    | 0.66                 | 0.45            |
> | ViT-B-16-text-plus    | DataComp | 128M    | 0.65                 | 0.43            |
> | ViT-B-32              | Relaion  | 128M    | 0.61                 | 0.38            |
> | ViT-B-32              | DataComp | 128M    | 0.55                 | 0.37            |
> | ViT-B-16-text-plus    | DataComp | 64M     | 0.55                 | 0.36            |
> | ViT-B-32              | OVid     | 30M     | 0.54                 | 0.31            |
> | ViT-B-32              | Relaion  | 64M     | 0.51                 | 0.30            |
> | ViT-B-32              | DataComp | 64M     | 0.44                 | 0.29            |
> | ViT-B-16-text-plus    | DataComp | 30M     | 0.38                 | 0.26            |
> | ViT-B-32              | Relaion  | 30M     | 0.37                 | 0.22            |
> | ViT-B-32              | DataComp | 30M     | 0.32                 | 0.21            |

---

### Official Review · Reviewer_c4tE · 2025-10-31

**Soundness:** 2
**Presentation:** 3
**Contribution:** 2
**Rating:** 6
**Confidence:** 2

**Summary:**

This paper introduces OVID, a massive open dataset comprising 10 million hours of video, positioning it as a scalable source for image-text data. The authors demonstrate that frames extracted from these videos, paired with machine-generated captions, constitute an image-text dataset with a distribution distinct from existing web-crawled image collections. They validate this by training CLIP models in image-text retrieval tasks.

**Strengths:**

- Unprecedented Scale: 10 million hours of video and 300 million frame-caption pairs.
- Notable Diversity: The dataset exhibits strong diversity across topics, languages, and video lengths, supporting a wide range of potential training scenarios.

**Weaknesses:**

1. Potential Circularity in Evaluation: The reliance on CLIP-based metrics (CLIPScore) for captioner selection and CLIP-based training for downstream validation may introduce a bias towards captions that align with the CLIP embedding space, rather than capturing broader semantic or human-aligned quality.
2. Insufficient Evidence for Distributional Difference: The central claim that video frames constitute a distributionally distinct and high-quality source of image-text data is only partially supported. The evidence rests heavily on the downstream performance of CLIP models, particularly on retrieval tasks. A more direct analysis—for instance, quantifying the domain shift between OVID frames and existing image corpora (e.g., using FID or other divergence measures)—would substantially strengthen this claim.
3. Limited Validation for Video Tasks: As acknowledged in the limitations, the dataset does not consider temporal or audio-visual aspects. While the frame-level data is validated for image-text tasks, the potential of OVID for video-language modeling (e.g., training ViCLIP or other video-text models) remains unexplored. This significantly reduces the demonstrated applicability of what is, fundamentally, a video dataset.

**Questions:**

- Could you provide a more quantitative or qualitative analysis of how the visual distribution of OVID frames differs from that of web-crawled image datasets?

---

> ### Author Response · Authors · 2025-11-25
> **Rebuttal (Part 1)**
>
> We thank the reviewer for their time and constructive feedback. We appreciate that the review thinks our dataset will make a wide-range of training scenarios possible. We will address the reviews’ concerns below:
> > 1. Potential Circularity in Evaluation: The reliance on CLIP-based metrics (CLIPScore) for captioner selection and CLIP-based training for downstream validation may introduce a bias towards captions that align with the CLIP embedding space, rather than capturing broader semantic or human-aligned quality.
>
> We thank the reviewer for raising this point. Prior work [1, 2, 3] has shown that CLIPScore is highly correlated with human caption preferences and with downstream zero-shot ImageNet performance of models trained on those captions. Since we generate new captions without available human references, CLIPScore provides a practical, reference-free metric applicable to arbitrary image-text pairs.
> Additionally, we did not rely solely on CLIPScore: we also selected captioner candidates using the OpenVLM Leaderboard, which ranks multimodal LLMs on standardized, human-relevant benchmarks. This aligns with established practices in recent synthetic captioning work [4, 5].
> We therefore believe the evaluation is well-grounded and avoids circularity in practice.
>
> [1] Nguyen, Thao, et al. "Improving multimodal datasets with image captioning." Advances in neural information processing systems 36 (2023): 22047-22069.
> [2] Kasai, Jungo, et al. "Transparent human evaluation for image captioning." Proceedings of the 2022 Conference of the North American Chapter of the Association for Computational Linguistics: Human Language Technologies. 2022.
> [3] Xu, Hu, et al. "Altogether: Image captioning via re-aligning alt-text." Proceedings of the 2024 Conference on Empirical Methods in Natural Language Processing. 2024.
> [4] Lai, Zhengfeng, et al. "Veclip: Improving clip training via visual-enriched captions." European Conference on Computer Vision. Cham: Springer Nature Switzerland, 2024.
> [5]Li, Xianhang, et al. "What If We Recaption Billions of Web Images with LLaMA-3?." arXiv preprint arXiv:2406.08478 (2024).
>
> > 2. Insufficient Evidence for Distributional Difference: The central claim that video frames constitute a distributionally distinct and high-quality source of image-text data is only partially supported. The evidence rests heavily on the downstream performance of CLIP models, particularly on retrieval tasks. A more direct analysis – for instance, quantifying the domain shift between OVID frames and existing image corpora (e.g., using FID or other divergence measures) -- would substantially strengthen this claim.
> > - Could you provide a more quantitative or qualitative analysis of how the visual distribution of OVID frames differs from that of web-crawled image datasets?
>
> We agree with the reviewer that we did not provide enough evidence that there exists a distribution shift. Therefore, we performed a FID analysis over CLIP embeddings (CLIP-ViT-B-32) from 100k randomly sampled images/frames from ReLAION and OVid. These are the results:
> - 100k ReLAION samples vs 100k Ovid samples: **33.9231**
> - 100k ReLAION samples vs another 100k ReLAION samples: **0.1581**
>
> This analysis indicates a substantial distribution shift between the OVID frame corpus and ReLAION. This shift suggests that OVID serves as a complementary data source to standard web-image collections for vision-language pretraining. We appreciate the reviewer’s suggestion and incorporated this result into the revised manuscript (Section 3.3).

---

> ### Author Response · Authors · 2025-11-25
> **Rebuttal (Part 2)**
>
> > Limited Validation for Video Tasks: As acknowledged in the limitations, the dataset does not consider temporal or audio-visual aspects. While the frame-level data is validated for image-text tasks, the potential of OVID for video-language modeling (e.g., training ViCLIP or other video-text models) remains unexplored. This significantly reduces the demonstrated applicability of what is, fundamentally, a video dataset.
>
> We agree that training and evaluating on video-specific tasks such as temporal localization or action recognition would be valuable future work. However, such training requires additional compute resources -- larger compared to reference CLIP model training as performed for validation in our work -- which was beyond compute budget available for the scope of this dataset release paper. This work validates the dataset by using paired video frame and text data from recaptioning to train reference CLIP models and obtain scaling trends for classification and retrieval as downstream tasks. This demonstrates potential of video data as pool for composing image-text datasets, complementing standard web crawling from CommonCrawl. This is important, as web crawling increasingly faces problems with synthetically generated data, which videos from content controlled platforms as we have in our work do not face yet.  Training large-scale video models on the other hand is an exciting next step, and we hope that the availability of OVid further enables such work by the community.

---

### Official Review · Reviewer_EeLg · 2025-11-01

**Soundness:** 2
**Presentation:** 2
**Contribution:** 2
**Rating:** 2
**Confidence:** 3

**Summary:**

The paper introduces OVID, an open large-scale video dataset designed to serve as a new source of high-quality image-text pairs. OVID comprises 10 million hours of video sourced from CommonCrawl, yielding 300 million captioned frames and 12 million video-level summaries generated through an automated captioning pipeline using the DeepSeek-VL2-tiny model. The authors train CLIP models on OVID and benchmark them against models trained on DataComp and Re-LAION, showing that OVID achieves state-of-the-art retrieval performance but slightly lower zero-shot classification accuracy on ImageNet. They argue that video-derived frames offer a novel and complementary data distribution to traditional web images. By releasing OVID and its curation pipeline openly, the work aims to democratize multimodal research and facilitate the development of open foundation models.

**Strengths:**

- This paper collected a new dataset with a large number of image and text pairs.

**Weaknesses:**

- The academic impact and the importance of the dataset are unclear. In Table 1, the authors only mention the number of samples. However, in terms of the number of samples, the advantage compared to the CoYo-300M dataset with 300M samples is unclear. The only difference is the source of the samples. However, the importance of the source is not discussed enough. A more detailed analysis of the diversity or bias to demonstrate the advantages of the proposed dataset is required.
-Overall, the advantage of the proposed dataset is unclear. In the experiment in Table 3, the authors only performed an engineering to find a favorable combination of existing models without any novel insight. In addition, the combinations covered in the experiment are not comprehensive enough.
- In Table 4, the experimental setting is questionable. The number of samples considered in the experiment is not consistent and not comprehensive. More importantly, CLIP models trained on OVID show weaker zero-shot classification performance, indicating limited fine-grained label alignment.
- Evaluation is restricted to retrieval and classification, offering limited evidence of general downstream utility. The authors should have considered other vision-language tasks such as image captioning.

**Questions:**

Please refer to the questions in the weakness section.

---

> ### Author Response · Authors · 2025-11-25
> **Rebuttal (Part 1)**
>
> We thank the reviewer for their time and feedback. We would like to emphasize that our contribution goes beyond the image-text pairs: we provide both the video URLs and the raw videos themselves (accessible to research institutions), which constitute the core of our dataset that can be used for various video modelling studies. The extracted image–text pair subset serves as a validation resource for the dataset and represents just one of many potential usage scenarios.
> > The academic impact and the importance of the dataset are unclear. In Table 1, the authors only mention the number of samples. However, in terms of the number of samples, the advantage compared to the CoYo-300M dataset with 300M samples is unclear. The only difference is the source of the samples. However, the importance of the source is not discussed enough. A more detailed analysis of the diversity or bias to demonstrate the advantages of the proposed dataset is required. -Overall, the advantage of the proposed dataset is unclear.
>
> The 300M subset we captioned and released on Hugging Face is not the core contribution; it merely serves to evaluate the quality of our video data as a source of image-caption pairs and provide one possible validation of its quality. The actual dataset contains ~10M video hours, corresponding to up to ~1T extractable unfiltered frames, orders of magnitude larger than CoYo-300M or any existing open dataset. While our validation experiments use a 300M-frame subset with generated captions, this approach naturally provides a scalable source of image-text pairs. Under a conservative estimate of the number of extractable keyframes from the full 1T-frame pool, the dataset can be extended to 2.6B image-text pairs using the same or an improved caption-generation pipeline. We will update the manuscript to clarify this point and highlight the potential scale.
>
> > In the experiment in Table 3, the authors only performed an engineering to find a favorable combination of existing models without any novel insight. In addition, the combinations covered in the experiment are not comprehensive enough.
>
> Table 3 reports our recaptioning ablation, where we systematically compare different language models as re-captioners in terms of model size, CLIPScores and throughput. Carefully ablating captioning models provides insight into how caption quality depends on the underlying LLM, which is a technically relevant and meaningful contribution. However, we want to emphasise that this is not the core contribution of this work but used for assessing the quality of the dataset as image-text source.

---

> ### Author Response · Authors · 2025-11-25
> **Rebuttal (Part 2)**
>
> > In Table 4, the experimental setting is questionable. … More importantly, CLIP models trained on OVID show weaker zero-shot classification performance, indicating limited fine-grained label alignment.
>
>
> We agree that the performance gap between classification and retrieval tasks deserves deeper analysis. This discrepancy could stem either from properties of our video dataset, e.g. inducing a certain image distribution, or from limitations inherent to synthetic captions. To disentangle the two, we have preliminary evidence reported in our paper comparing performance on recaptioned DataComp (Tab. 4) vs recaptioned OVid (Tab. 5) subset. We observe similar gaps for DataComp, e.g. on 128M sample seen scale, we obtain 0.40 (original DataComp) vs 0.23 (recaptioned DataComp) for 0-shot classification IN1k, while for 0-shot retrieval on MS-Coco, we get 0.33 (original DataComp) vs 0.42 (recaptioned DataComp). We provide an overview in the table below.
>
> | Training data     |   IN1K classification  |   MSCOCO retrieval  |
> |:------------------|:----------------------:|:-------------------:|
> | DataComp          |           0.40         |  0.33               |
> | DataComp, recapt. |           0.23         |  0.42               |
> | OVid, recapt.  |           0.19         |  0.48               |
>
>
> We see thus that recaptioning induces the same pattern – drop in classification but increase in retrieval performance – on a different reference dataset (DataComp) that is known for yielding strong IN1K classification performance in its original form with alt text captions. Therefore, we can conclude that not image distribution, but captions are the key factor changing the dataset behavior and resulting in different trends on classification and retrieval. We see on the other hand that on retrieval, recaptioned OVid is outperforming recaptioned DataComp using the same recaptioning procedure, showing that while recaptioning alone boosts retrieval, image distribution induced by extracting frames from Ovid benefits from the recaptioning even more strongly. Prior work (e.g.,[1]) has also shown that synthetic captions typically underperform on fine-grained classification tasks but excel in retrieval, which is consistent with our observations. We will update the manuscript with corresponding content and clarifications.
>
> We additionally analyze the caption-length distributions and the frequency of ImageNet-1k (IN1k) class names in the 300M Ovid-recaptioned subset and the 300M DataComp subset. Our analysis shows that Ovid captions have a much narrower length distribution, with longer captions dominating, whereas DataComp captions exhibit a broader distribution with a clear prevalence of short captions. More importantly, we observe only 21M IN1k class-name occurrences in the Ovid captions, compared to 141M in DataComp (see https://imgur.com/a/TOnOQ6L). This gap provides a clear explanation for the weaker IN1k zero-shot classification performance of Ovid-trained models: real captions in DataComp align much more strongly with the IN1k class structure. Consequently, adapting the caption-generation pipeline to better capture IN1k class names is likely to improve performance on IN1k classification. We further assess the OVid-trained CLIP models on Flickr30k retrieval, a setting more aligned with our synthetic captions, and also observe strong performance (see Table 1). We have added this analysis to Appendix B in the revised manuscript.
>
> [1]: https://arxiv.org/pdf/2310.07699
>
> > … The number of samples considered in the experiment is not consistent and not comprehensive. …
>
> We use 300M extracted frames from the 10M video hours of downloaded videos with minimal filtering (back-scene detection and scene changing frames). This is used for subsequent tests on CLIP models and scaling laws. Could the reviewer specify where they think there is an inconsistency? This would help us to improve the clarity of the manuscript.
>
> > Evaluation is restricted to retrieval and classification, offering limited evidence of general downstream utility. The authors should have considered other vision-language tasks such as image captioning.
>
>
> We agree that classification and retrieval cover only a subset of possible downstream tasks. However, both tasks directly probe the richness and generality of the learned embedding space across a wide range of objects, scenes, and visual concepts. Strong performance in these settings is a well-established indicator of data diversity and overall dataset quality. While broader downstream evaluations are possible, these two tasks already provide meaningful and reliable evidence for the utility of our video dataset as a source of image-text training data.

---

> ### Author Response · Authors · 2025-11-25
> **Table 1: Image-text retrieval performance (R@5) on Flickr30k and MSCOCO for models trained on different 300M–30M subsets of OVid, DataComp, and Relaion. Scaling laws for OVid persist on different models.**
>
> | Model                  | Dataset   | Samples | Flickr30k I→T R@5 | MSCOCO I→T R@5 |
> |-----------------------|----------|---------|----------------------|-----------------|
> | ViT-B-16-text-plus    | OVid     | 300M    | 0.87                 | 0.63            |
> | ViT-B-16-text-plus    | OVid     | 128M    | 0.82                 | 0.56            |
> | ViT-B-32              | OVid     | 300M    | 0.82                 | 0.56            |
> | ViT-B-16-text-plus    | Relaion  | 300M    | 0.80                 | 0.53            |
> | ViT-B-16-text-plus    | OVid     | 64M     | 0.79                 | 0.50            |
> | ViT-B-16-text-plus    | DataComp | 300M    | 0.75                 | 0.52            |
> | ViT-B-32              | OVid     | 128M    | 0.75                 | 0.48            |
> | ViT-B-32              | Relaion  | 300M    | 0.72                 | 0.46            |
> | ViT-B-16-text-plus    | Relaion  | 128M    | 0.71                 | 0.45            |
> | ViT-B-16-text-plus    | OVid     | 30M     | 0.67                 | 0.40            |
> | ViT-B-32              | OVid     | 64M     | 0.67                 | 0.40            |
> | ViT-B-32              | DataComp | 300M    | 0.66                 | 0.45            |
> | ViT-B-16-text-plus    | DataComp | 128M    | 0.65                 | 0.43            |
> | ViT-B-32              | Relaion  | 128M    | 0.61                 | 0.38            |
> | ViT-B-32              | DataComp | 128M    | 0.55                 | 0.37            |
> | ViT-B-16-text-plus    | DataComp | 64M     | 0.55                 | 0.36            |
> | ViT-B-32              | OVid     | 30M     | 0.54                 | 0.31            |
> | ViT-B-32              | Relaion  | 64M     | 0.51                 | 0.30            |
> | ViT-B-32              | DataComp | 64M     | 0.44                 | 0.29            |
> | ViT-B-16-text-plus    | DataComp | 30M     | 0.38                 | 0.26            |
> | ViT-B-32              | Relaion  | 30M     | 0.37                 | 0.22            |
> | ViT-B-32              | DataComp | 30M     | 0.32                 | 0.21            |

---

### Comment · Area_Chair_ztkm · 2025-11-26
**A Reminder on Your Crucial Role in the ICLR Discussion Period**

Dear Reviewers who haven't engaged with the rebuttal:

As the Area Chair, I would like to sincerely thank you for the time and expertise you have invested in writing your initial review. Your insights are invaluable to the decision-making process.

We are now entering the critical discussion and rebuttal phase. This is a collaborative process where authors have the opportunity to address your concerns and questions. Your active participation in this phase is essential to ensure we reach a fair and well-informed final decision.

I strongly encourage you to:

Engage with the Authors' Rebuttal: Please read the authors' response carefully and substantively.

Participate in the Discussion: Engage with the other reviewers on the forum. If the authors have clarified a point, please acknowledge it. If you have follow-up questions or remaining concerns, please voice them. Your dialogue with fellow reviewers is key to reaching a consensus.

Update Your Review (if necessary): Based on the discussion and rebuttal, you may feel the need to adjust your score or final recommendation. Please do so, as it reflects a more holistic view of the paper.

Your continued engagement ensures the integrity and quality of the ICLR conference. Thank you for your vital contribution to our community.

Best regards,

Area Chair, ICLR 2026

---

### Note · Authors · 2026-06-17

I have read and agree with the venue's withdrawal policy on behalf of myself and my co-authors.

---

### Meta-Review · Area_Chair_q4jm · 2026-01-06

**Summary:**

This paper presents a large-scale video-frame-caption dataset designed to serve as a novel source of high-quality image-text pairs and facilitate the training of CLIP models at multiple scales. The initial ratings were 8/6/2/2 and the score remained unchanged before the reset.

The reviewers’ primary concerns initially centered on the **unclear significance and impact** of the dataset, the **insufficient evaluation scope** (specifically the lack of video-specific and downstream task validation), and the **notable performance gap** in zero-shot classification compared to established web-based baselines like Re-LAION. Furthermore, reviewers highlighted a lack of rigorous data analysis regarding **distributional shifts, diversity, and potential biases**.  In the rebuttal, the authors sought to address these concerns by providing **FID analysis** to quantify domain differences and attributing the zero-shot performance lag to the **verbose nature of synthetic captions**. They also justified the omission of video benchmarks by citing **computational constraints** and offered clarifications on methodological choices.

Despite the extensive rebuttal, several critical issues remain unresolved. Reviewer **8Gxi** underscored that the dataset's **practical utility**—particularly whether it yields empirical gains over training on web data alone—remains unsubstantiated, as the authors provided no evidence of joint training or superior performance compared to web-only baselines. Furthermore, the reviewer emphasized that downstream video experiments are essential for validating video-sourced representations, a core requirement that the authors failed to address in their response.

After a careful assessment of the submission, reviews, response, and discussion, the AC recommends rejection. The authors are encouraged to revise and refine the manuscript in accordance with the reviewers’ feedback for a future submission.

**Reviewer Concerns:**

The reviewers’ primary concerns initially centered on the **unclear significance and impact** of the dataset, the **insufficient evaluation scope** (specifically the lack of video-specific and downstream task validation), and the **notable performance gap** in zero-shot classification compared to established web-based baselines like Re-LAION. Furthermore, reviewers highlighted a lack of rigorous data analysis regarding **distributional shifts, diversity, and potential biases**.  In the rebuttal, the authors sought to address these concerns by providing **FID analysis** to quantify domain differences and attributing the zero-shot performance lag to the **verbose nature of synthetic captions**. They also justified the omission of video benchmarks by citing **computational constraints** and offered clarifications on methodological choices.

Despite the extensive rebuttal, several critical issues remain unresolved. Reviewer **8Gxi** underscored that the dataset's **practical utility**—particularly whether it yields empirical gains over training on web data alone—remains unsubstantiated, as the authors provided no evidence of joint training or superior performance compared to web-only baselines. Furthermore, the reviewer emphasized that downstream video experiments are essential for validating video-sourced representations, a core requirement that the authors failed to address in their response.

**Reviewer Scores:**

The manuscript received initial review scores of 8/6/2/2. After the rebuttal/discussion and before the reset, the score remained unchanged.

Since several concerns raised by the reviewers may remain unresolved after the rebuttal (see 'Reviewer Concerns'), I would approximate 8/6/2/2 as the final score.

---

### Decision · Program_Chairs · 2026-01-26

Reject